



# Hyperfine-Resolution Mapping of On-Road Vehicle Emissions with Comprehensive Traffic Monitoring and Intelligent Transportation System

Linhui Jiang[1], Yan Xia[1], Lu Wang[1], Xue Chen[1], Jianjie Ye[5], Tangyan Hou[1], Liqiang Wang[1], Yibo Zhang[1], Mengying Li[1], Zhen Li[1], Zhe Song[1], Yaping Jiang[1], Weiping Liu[1], Pengfei Li[3+], Daniel Rosenfeld[4], John H. Seinfeld[2], Shaocai Yu[1,2+]

[1]Research Center for Air Pollution and Health; Key Laboratory of Environmental Remediation and Ecological Health, Ministry of Education, College of Environment and Resource Sciences, Zhejiang University, Hangzhou, Zhejiang 310058, P.R. China
[2]Division of Chemistry and Chemical Engineering, California Institute of Technology, Pasadena, CA 91125, USA.
[3]College of Science and Technology, Hebei Agricultural University, Baoding, Hebei 071000, P.R. China
[4]Institute of Earth Sciences, The Hebrew University of Jerusalem, Jerusalem, Israel
[5]Bytedance Inc., Hangzhou, Zhejiang 310058, China

[+]*Correspondence to*: Shaocai Yu (shaocaiyu@zju.edu.cn); Pengfei Li (lpf_zju@163.com)



## Abstract

Urban on-road vehicle emissions affect air quality and human health locally and globally. Such emissions typically exhibit distinct spatial heterogeneity, varying sharply over short distances (10 m ~ 1km). However, all-around observational constraints on the emission sources are limited in much of the world. Consequently, traditional emission inventories lack the spatial resolution that can characterize on-road vehicle emission hotspots. Here we establish a bottom-up approach to reveal a unique pattern of urban on-road vehicle emissions at 1 ~ 3 orders of magnitude higher spatial resolution than current inventories. We interconnect all-around traffic monitoring (including traffic fluxes, vehicle-specific categories, and speeds) via an intelligent transportation system (ITS) over the Xiaoshan District in the Yangtze River Delta (YRD) region. This enables us to calculate single-vehicle-specific emissions over each fine-scale (10 m ~ 1 km) road segment. Thus, a hyperfine emission dataset is achieved, and on-road emission hotspots appear. The resulting map shows that the hourly average on-road vehicle emissions of CO, $NO_x$, HC, and $PM_{2.5}$ are 74.01 kg, 40.35 kg, 8.13 kg, and 1.68 kg, respectively. More importantly, widespread and persistent emission hotspots emerge, of significantly sharp small-scale variability, up to 8 ~ 15 times, attributable to distinct traffic fluxes, road conditions, and vehicle categories. On this basis, we investigate the effectiveness of routine traffic control strategies on on-road vehicle emission mitigation. Our results have important implications for how the strategies should be designed and optimized. Integrating our traffic-monitoring-based approach with urban air quality measurements, we could address major data gaps between urban air pollutant emissions and concentrations.





## 1. Introduction

Rapid growth in vehicle population for decades has led to widespread and severe levels of fine particulate matter ($PM_{2.5}$) and ozone ($O_3$) (Anenberg et al., 2017; He et al., 2020; Huang et al., 2020; Kelly and Zhu, 2016; Tessum et al., 2014; Zhang et al., 2012, 2019). However, the gradients of on-road vehicle emissions are not well represented in routine emission inventories. That is, the traffic states (e.g., traffic fluxes, road conditions, and vehicle categories) can vary sharply over short distances (10 m ~ 1

km), particularly in urban zones (Chen et al., 2020; Gately et al., 2017; Liu et al., 2019; Wu et al., 2019; Yu et al., 2020). Routine inventories of on-road vehicle emissions are established based on macro-scale and retrospective statistics. Consequently, they are temporally static for a historical year or month and spatially coarse (> 1 ~ 25 km) (Janssens-Maenhout et al., 2015; Li et al., 2017; Zhang et al., 2013). Earlier studies have applied traffic models to improve spatiotemporal resolution (Zhang et al., 2016). However,

given that the traffic states were assumed, the simulated emissions were prone to deviate from real-world situations, especially from fine-scale gradients. More importantly, the emission hotspots and their anthropogenic drivers are missed.

     Recently, significant advances have been made in comprehensive traffic monitoring techniques. These methods include GPS-instrumented floating cars (e.g., GPS-equipped probe taxis), open-access

congestion maps, radio frequency identification, and traffic video records, each of which has distinct advantages and limitations (Gately et al., 2017; Gately and Hutyra, 2017; Jing et al., 2016; Liu et al., 2018; Wen et al., 2020; Wu et al., 2019; Yang et al., 2019, 2018b). The individual GPS-instrumented floating cars allow us to extrapolate regional-scale vehicle activity levels. Yet, they are relatively scarce compared to the whole fleet, unable to characterize the fine-scale gradients (10m ~ 1km) as well as emission hotspots.

Open-access congestion maps typically originate from navigation software, such as Baidu Map. Technically, they collect locations of individual mobile phones as real-time traffic information. On this basis, hierarchical traffic congestion indices can be built up and treated as spatiotemporal surrogates of traffic fluxes and speeds. Despite this, the information for individual vehicles, like speed and categories, remains unavailable. A recent study (Deng et al., 2020) utilized the BeiDou Navigation Satellite System

to develop a full-sample high-resolution emission inventory, but only for trucks.





In contrast, comprehensive traffic monitoring technologies, such as radio frequency identification (Paul et al., 2013) coupled with traffic video records (Song et al., 2019), can offer valuable opportunities to obtain real-time vehicle-specific traffic information. Despite this, in the United States, daily traffic activities are released annually at the state-level rather than at high spatiotemporal resolution (hourly and

10m ~ 1km) (Gately et al., 2013). On the other hand, those facilities are inter-complementary but usually owned and operated by different governmental agencies or private companies. Hence, relying on either party, hyperfine-resolution emission inventories cannot be derived comprehensively.

It is essential to introduce an intelligent transportation system (ITS) that is capable of interconnecting independent traffic monitoring and thus offering a complete picture of traffic states (Avila

and Mezić, 2020; Yang et al., 2020; Zhang et al., 2018). Here, we derive a hyperfine-resolution on-road vehicle emission inventory. For most developing regions, especially in populous parts of Asia and Africa, such an integrated system is largely absent.

As one of the most developed regions in the Yangtze River Delta (YRD), the Xiaoshan District is confronting severe air pollution, particularly with surface $O_3$ frequently exceeding air quality standards

in summertime (http://www.cnemc.cn/). Moreover, this is one of few areas, in which comprehensive traffic monitoring realizes full coverage and is interconnected via an ITS (named "City Brain") since 2017 (Fig. 1) (Hua, 2018). This allows one to calculate single-vehicle-specific emissions over each fine-scale (10 m ~ 1 km) road segment. Consequently, here we derive the largest and most hyperfine urban on-road vehicle emission dataset of its type. Thus, fine-scale gradients (10 m ~ 1 km) and hotspots of on-road

vehicle emissions are exposed. On this basis, we can directly evaluate the potential impacts of precise (e.g., vehicle-type-specific, road-segment-specific, or traffic-flux-specific) emission mitigation strategies. Our results provide new insights into the spatial variability of urban on-road vehicle emissions.

## 2. Materials and methods

The objective of this study is to apply a bottom-up model approach to establish a hyperfine-

resolution inventory of on-road vehicle emissions over the Xiaoshan District in the YRD (Fig. 1). All key input data, including traffic fluxes, vehicle-specific categories, and vehicle speeds, were obtained from comprehensive traffic monitoring coupled with an ITS. Besides, vehicle-specific emission factors come



from the local official vehicle Inspect/Maintenance (I/M) dataset, the methodology of which has been described in China's National Emission Inventory Guidebook (ICCT, 2020).

### 2.1. Comprehensive traffic monitoring network

The Xiaoshan District, located in the YRD in China (Fig. 2) had in 2019 a population of over 1.58 million, 18.56% of the population of New York City. Its GDP was close to 200 billion Yuan, ranking fifth among districts in China. The total length of the road network was around 2000 km within a limited geographical extent (i.e., 1417.83 $km^2$). With this background, the Xiaoshan District has become an important urban transportation hub in the YRD, for which on-road vehicle emissions were projected to be intensive and to play a critical role in affecting fine-scale air quality (and exposure equity). Since 2016, routine measures to ease traffic congestion, such as license restrictions during the morning and afternoon rush hours, were implemented over the Xiaoshan District. This would significantly alter fine-scale spatiotemporal patterns of traffic states (including traffic fluxes, vehicle speeds, and fleet compositions) and thus on-road vehicle emissions, the impacts of which remain unclear.

### 2.2. Hyperfine-resolution bottom-up model framework

A hyperfine-resolution bottom-up model framework was established to calculate primary on-road vehicle emissions, including carbon monoxide (CO), hydrocarbons (HC), nitrogen oxides ($NO_x$), and $PM_{2.5}$. Figure 1 is a flow diagram to illustrate the overall methodology for this framework. The results depend on an ensemble calculation of traffic fluxes, road segments, vehicle-specific speed, categories, and emission factors (Eq. 1) (Wu et al., 2019; Yang et al., 2019; Zhang et al., 2016):

$$E_{h,j,l} = \sum_t EF_{c,j}(v) \times TF_{c,h,l} \times L_l, \text{ (Eq. 1)}.$$

$E_{h,j,l}$ is the consequent emission of the pollutant $j$ on the road link $l$ at the hour $h$, the unit of which is grams per hour (g h$^{-1}$), while the remaining variables denote the input data for the model. $EF_{c,j}(v)$ is the average emission factor of the pollutant $j$ for the vehicle category $c$ at the speed $v$, the unit of which is grams per kilometre (g km$^{-1}$); $TF_{c,h,l}$ is the traffic flux of the vehicle category $c$ on the road segment $l$ at the hour $h$, in units of vehicles per hour (veh h$^{-1}$); $L_l$ is the length of the road segment $l$ in units of kilometres (km).



The major technical advance in this study is that all these input data were vehicle-specific and
obtained from all-round traffic monitoring, introduced and detailed in Sect. 2.3. Detailed road segments
are important carriers reflecting traffic states and on-road vehicle emissions. In this study, we divided the
entire road network over the Xiaoshan District into 1894 road segments. Over the entire district, such
road segments were divided into three road classes: highways, arterial roads, and residential streets (Fig.
2). Spatially, each road segment was adaptive to a set of traffic monitoring platforms that can collect
comprehensive traffic profiles, including traffic fluxes, vehicle-specific categories, and speeds. Therefore,
the all-round traffic monitoring derived the hyperfine-resolution map of road segments and thus $E_{h,j,l}$.

To obtain the key input data for this model, we explored traffic monitoring information that
achieved full coverage over the Xiaoshan District (Fig. 2). On this basis, vehicle-specific speed and
categories were collected. Besides, the vehicle-category-specific emission factors were obtained from the
local official I/M dataset. Consequently, an ITS (named "City Brain") was developed to simultaneously
upload the traffic monitoring information and, more importantly, to establish vehicle-specific links
between those parameters. Therein, traffic fluxes played a major role in affecting vehicle emissions (Deng
et al., 2020; Yang et al., 2019). Particularly, traffic congestion in narrow spaces might contribute to urban
emission hotspots. To this end, traffic video records, together with image recognition algorithms, were
applied to detect vehicle license plates and thus to monitor traffic fluxes. As a result, we constructed a
total dataset of 254.31 million records from 13 November 2020 to 13 January 2021. Such measurements
were recorded by different video facilities integrated into the ITS and thus accessible simultaneously.
Moreover, accurate vehicle speeds are another key driver that is of great significance for optimizing
vehicle emission factors (Yang et al., 2019). Here, together with traffic fluxes, the vehicle-specific speed
was measured concurrently by radar velocimeters. Collectively, a high-resolution map of traffic states,
including traffic fluxes and vehicle-specific speed, was captured by comprehensive traffic monitoring
over the Xiaoshan District. From this perspective, this work is distinct from previous attempts that
introduced spatial surrogates for traffic states (e.g., floating cars and traffic congestion index in open-
source maps) to fill the monitoring gaps. Other key input data are the vehicle-specific category that is
closely related to vehicle-specific emission factors (Huang et al., 2020). For each vehicle, an image
detection technology based on traffic video records was utilized to recognize license plate and category.



According to the license plate, the identified vehicle category is verified via the I/M data. Herein, six vehicle categories were detected and defined, including light-duty vehicles (LDVs), middle-duty vehicles (MDVs), heavy-duty vehicles (HDVs), light-duty trucks (LDTs), middle-duty trucks (MDTs), and heavy-duty trucks (HDTs).

According to the detected license plates, these vehicles fell into two types: registered vehicles and non-registered ones in the local official I/M dataset. The emission factors of the former were vehicle-category-specific and speed-dependent, obtained from the I/M dataset, while those of the latter were also speed-dependent but vehicle-category-specific averages (Fig. S1). Here, the detailed species profiles of HC, as well as evaporative HC, were not included.

### 2.3. Traffic control strategies

Based on a hyperfine-resolution map of on-road vehicle emissions, the impacts of traffic control measures on vehicle emission reductions can be directly investigated. Here, four scenarios were designed, which were mainly oriented to traffic fluxes and fleet compositions. Therein the key point was to determine how to conduct these strategies spatially and temporally (Table 1). First, the routine scenario (S1) was conducted during the morning and evening rush hours (from 7:00 to 9:00 and from 16:30 to 18:30, Local Time) on weekdays (i.e., from Monday to Friday). It required that vehicles with specific tail numbers of the license plates were prohibited on the arterial and residential roads. For instance, the prohibited tail numbers were 1 and 9 on Monday. Other detailed rules were illustrated in Table 1. Second, similar to but more stringent than the routine scenario (S1), the scenario (S2) adopted the even-odd rule to reduce the traffic fluxes in half at the same space. Third, the truck scenario (S3) oriented at both local registered and non-registered trucks, which were strictly prohibited all day long over the highways. Finally, the G20 scenario (S4) reflected the traffic states during the G20 summit in 2016 over the Xiaoshan District, with much stricter traffic limitations than normal situations (Ji et al., 2018; Wang et al., 2020; Zhang et al., 2020). It can be regarded as the combination of the scenario (S2) and the truck scenario (S3). Thus, all vehicles should comply with the even-odd rule over the entire district.





### 2.4. Monte Carlo subsampling

The hyperfine bottom-up model was a big-data-driven framework. We should thus apply a sub-sampling analysis to evaluate the stability of the resulting on-road vehicle emission inventory. The objective was to investigate the extent to which less repeated traffic monitoring information can reproduce the long-term spatial emission patterns driven by the entire dataset (Apte et al., 2017; Hankey and Marshall, 2015).

Here we focus only on weekdays rather than weekends when the expected casual trips might affect the analysis. We utilized Monte Carlo simulations to subsample the full-traffic-monitoring-driven emissions repeatedly. Briefly, we randomly sampled the emission information in unique weekdays ($1 \leqslant N \leqslant 42$) at each road segment from our universal dataset. Each road segment has 35 weekdays of sampling on average. For each value of **N**, we performed 1000 random draws to generate 10000 subsampled "maps" of fine-scale hourly average emissions. For road segments with fewer than **N** days of sampling, the "subsampling" effectively contained all data. Consequently, the subsampled maps converged to the full-data-driven results, since **N** approached the total number of full traffic monitoring information.

We adopted three metrics to compare the performance of each subsampled emission map to that of the full-data-driven result. First, as a metric of precision, we calculated the $\gamma^2$ for them. The second metric was the normalized root-mean-square error (i.e., the coefficient of variation of the RMSE, CVRMSE). Third, as a metric of the temporal stability of sub-sampled spatial patterns, we calculated the intraclass correlation (ICC) of each sub-sampled iteration, grouped by road segments. The ICC is a metric that results from one-way Analysis of Variance (ANOVA) to quantify the degree of similarity among repeated measurements within individual groups (i.e., road segments). After computing the ANOVA for data grouped by road segments, the ICC is calculated as the ratio of the variability between groups (Mean Squares of Treatment/Group, MST) to the sum of the MST and the variability within individual groups (Mean Square Error, MSE):

$$ICC = \frac{MST}{MST+MSE} \text{ (Eq. 2).}$$



By definition, ICC is bounded from 0 to 1. For a hypothetical dataset where all repeated measurements at each location were precisely equal to each other, the ICC would converge to 1.0. In contrast, for a dataset where the concentration variability among repeated measures at each individual location is very high relative to the spatial differences in concentration among roads, the ICC would approach 0. For this application, ICC values of 0.75 ~ 1 reflected large and systematic spatial differences, with low residual temporal variability at each location.

## 3 Results and discussion

### 3.1 Traffic characteristics and hotspots

Comprehensive traffic monitoring, coupled with the ITS, painted vivid pictures of within-urban traffic states, including traffic fluxes, fleet compositions, and traffic speeds (Figs. 3, 4 and Figs. S2 ~ S5). Remarkable spatiotemporal heterogeneities in fine-scale patterns were revealed. First, most (i.e., > 96.49%) of the traffic fluxes concentrated over the arterial roads and the residential streets rather than the highways (Fig. 3a). Figure 3b illustrates fine-scale variabilities in the traffic fluxes for an indicative ~ 1 $km^2$ urban zones. Within this small area, the hourly average traffic fluxes varied by more than 15 times. Even within individual roads, they still varied by more than eight times overall. An expected feature throughout the traffic monitoring dataset was the ubiquity of sharp spatial "traffic hotspots" (length < 100 m). Such hotspots were tentatively classified as individual road segments or clusters, where traffic fluxes exceeded the median level over the whole district. Figure 4 confirms potential causes for an indicative set of hotspots via imagery analysis. A uniform explanation was traffic congestion that, however, resulted from different drivers, such as large traffic fluxes in major arterial roads and their intersections or constructions in the middle of the roads. Such supplemental information provides further details on the hotspot identification scheme.

Second, the hourly average traffic fluxes on weekdays were close to those on weekends (Fig. 3 and Fig. S3). Nevertheless, the variation tendencies displayed a distinct picture during different moments between weekdays and weekends (Fig. S2). On weekdays, the diurnal traffic fluxes showed dramatic fluctuations, two peaks at 07:00 and 17:00, obviously related to the morning and evening rush hours. We



noted that such temporal peaks enhanced extensive spatial hotspots, spatially consistent with the above
hotspots based on the hourly average data but quantitatively more prominent (Fig. 3). This was because
that the morning and evening rush hours deteriorated the traffic congestion (Fig. 4). By comparison, the
variation extent on weekends was slightly lower than on weekdays and the early peak appeared two hours
later (Fig. S2). Overall, the maximum peak on weekends barely hit roughly 96.46% of those on weekdays.
Spatially, the hotspots of the traffic fluxes on weekdays were mostly consistent with those on weekends
but more variable, reflecting frequent casual travels (Fig. S3). Collectively, the fine-scale spatiotemporal
patterns of traffic fluxes over the entire district, particularly the hotspots, relied more on those on
weekdays.

Third, significantly strong correlations were found between the traffic speeds and fluxes spatially
and temporally. Following the traffic fluxes, the simultaneous vehicle-specific speeds fluctuated
substantially throughout the day (Fig. S2). When the traffic fluxes peaked at the morning and evening
rushes, the vehicle-specific speeds were expected to be at their lowest. Although the peaks changed from
weekdays to weekends, the valleys kept following such peaks. Spatially, the traffic flux hotspots likely
determined the traffic speed hotspots, particularly at the morning and evening rush hours (Fig. S4). In
contrast, the vehicle categories were independent of the traffic fluxes. Their diurnal variations showed
relative stability, even for different road types, after the morning rush hours (Fig. S2 and Fig. 3). On the
other hand, the HDVs and HDTs peaked in the early hours of the morning (i.e., from 1:00 to 5:00). Besides,
a striking picture lay in the spatial distributions (Fig. 3 and Fig. S5). The four types of vehicles, including
LDVs, MDVs, LDTs, and MDTs, flocked over (98.52%) the arterial and residential roads, while the rest
of vehicle categories, i.e., HDVs and HDTs, concentrated over (3.61%) the highways. Figure 3 details
fine-scale spatial distributions of HDVs and HDTs for three indicative highway zones (~ 1 km$^2$). Therein
the spatial hotspots scattered extensively. According to the image analysis (Fig. 4), the traffic congestion
attributed to the large traffic fluxes of HDVs and HDTs should be the unique driver. Therefore, the fleet
compositions would also affect emission distributions significantly, particularly over fine-scale zones.



## 3.2 Characteristics of on-road vehicle emissions

We established a hyperfine-resolution on-road vehicle emission inventory and captured the emission hotspots (Fig. 5 and Fig. S6). Overall, the hourly average emissions were summed up based on the classified roads (Table S1). It was clear that the emission intensities in the arterial roads, residential streets, and highways followed a descending order, although the residential streets were of the longest length and the largest traffic fluxes. The leading cause was the difference in vehicle categories in different

road types (Table S1). For instance, we estimated that the hourly average $NO_x$ emission intensities in the arterial roads, residential streets, and highways were 157.76 g/km, 135.22 g/km, and 107.83 g/km, respectively. For the highways, the hourly average traffic fluxes were 10277, accounting for 1.99% of the total amount, while their emissions amounted to more than 2.04 % of the total emissions.

        From the temporal perspective, on-road vehicle emissions of CO, HC, $NO_x$, and $PM_{2.5}$ showed

similar trends all day (Fig. S7). For instance, the daytime $NO_x$ emissions accounted for approximately 85.90% of the daily total emissions. Moreover, the $NO_x$ emissions varied throughout the day but reached agreement among the different road types (i.e., the arterial roads, residential streets, and highways). Yet, there was an apparent difference between emissions on weekdays and those on weekends. Similar to the temporal variations in the traffic fluxes on weekdays, those in on-road vehicle emissions also peaked at

the morning and evening rush hours. In turn, such temporal patterns were indistinct on weekends. Spatially, the high hourly average emissions with the unprecedented hyperfine resolution spread all over the district (Fig. 5 and Fig. S6). Such spatial pattern was distinct from previous results that generally show radiating decreases from the centre to the periphery (Jing et al., 2016; Yang et al., 2019), mostly associated with the spatial patterns of the traffic fluxes and vehicle categories (Fig. 3 and Fig. S4). Specifically, the

high emissions at the centre were mainly attributed to the high traffic fluxes and low traffic speeds. Note that, on the border of the district, the emission intensities in the residential streets far exceeded (> 436%) those in the neighbouring highways. This divergence could be interpreted by the spatial emission distributions of different vehicle categories (Fig. 3 and Fig. S5). For instance, the emissions of HDVs and HDTs in the residential streets contributed the most (79.79%), much higher than those (1.34%) in the

neighbouring highways.


### 3.4 Emission hotpots and drivers

The emissions of each pollutant (i.e., CO, HC, $NO_x$, and $PM_{2.5}$) generally peaked at the major road intersections, leading to the spatial emission hotspots (Fig. 5 and Fig. S6). The highest hourly average emissions occurred at the intersection of two arterial roads (i.e., North Shixin Road and Shanyin Road), where the measurement monitored the largest traffic fluxes (Fig. 3). At broader spatial scales, these hotspot emissions varied substantially among different road types. For instance, the hourly average emissions for the hotspots in Tonghui North Road and Hongda Road (i.e., arterial roads) were approximately consistent with those in Benjing Road and Hongni Road (i.e., residential streets) (Fig. 5 and Fig. S6, arterial roads vs. residential streets: 374.91g/km vs. 216.12g/km CO; 48.90g/km vs. 34.13g/km HC; 180.61g/km vs. 348.38g/km $NO_x$; 7.41 vs. 15.81g/km $PM_{2.5}$). In contrast, the hotspot emissions in the arterial roads and highways were substantially elevated above the residential levels. For CO, the hotspot emissions in the highways [arterial roads] exceed those on the residential roads by a factor of 2.1 [2.3]; for HC by a factor of 1.4 [2.6]; for $NO_x$ by a factor of 0.9 [1.1]; and for $PM_{2.5}$ by a factor of 0.8 [1]. Therein, emission hotspots for a given highway road were typically intensive and evident in several areas of the Xiaoshan District (Fig. 5). For instance, we estimated consistently higher (1.2 ~ 2 times) emission levels on a highway (i.e., Airport Road) than those on the neighbouring residential street (i.e., Wenming Road) (Fig. S8). More importantly, the diurnal emission hotspots remained stable spatially (Fig. S9), consistent with the hourly average level (Fig. 5 and Fig. S6). However, the emission intensities of the hotspots varied during different moments between weekdays and weekends (Fig. S10). The higher emissions of hotspots typically appeared at 08:00 and 18:00 on weekdays, 2.2 ~ 3.4 times larger than the hourly average level.

The indicative hotspots over the urban zones generally stretched for 100 ~ 200 m (Fig. 5). Over the short length of the transects, the hourly average emissions rose and fell more than 2.2 times. From the hotspot cores outwards, the hourly average emissions consistently followed "distance-decay" relationships (Fig. 6). An unconstrained three-parameter exponential model, $E(d) = \alpha + \beta \exp\left(-3d/k\right)$, reproduced the emission-distance relationship $E(d)$ with high fidelity ($r^2 = 0.96$). Specifically, the isotropic parameter $d$ reflected the distance to the hotspot cores (**m**); the background parameter $\alpha$ represented the background emissions far from the hotspots ($d \rightarrow 1000m$); the parameter $\beta$ represented



the emission increment resulting from proximity to the hotspots; the decay parameter $k$ governed the
spatial scale over which emissions relaxed to $\alpha$. For all pollutants, estimated values of $(\alpha + \beta)$
approached a constant (1.0), indicating that the combined contribution of the background and the near-
hotspot increment approached the hotspot emission levels.

Figure 6 shows the decay patterns of the hourly average emissions over urban zones on weekdays.
These results reflected the hourly average emission ratios (normalized at the hourly average emissions of
the hotspots) from hotspots outwards as a function of the distance ($d$). Note that the ratios of the hourly
average traffic fluxes and vehicle category proportions were calculated in the same way.

Consistent with expectations about the speed-dependent emission factors (Fig. S1), our estimated
distance-decay relationships were sharpest for $NO_x$, intermediate for HC and $PM_{2.5}$, and most shallow for
CO. In theory, comprehensive traffic profiles underlay the estimated hotspot emissions. To elucidate
determinants of the emission hotspot patterns, we mined those traffic data (Fig. 3, Fig. 5, and Fig. S6).
As expected, we found that the traffic fluxes largely shaped the spatial emission hotspot patterns over the
arterial and residential roads. Besides, the specific vehicle categories (i.e., HDVs and HDTs) also play a
key role.

For the emission hotspots in the highways, the traffic fluxes and emissions were both distance-
dependent and applicable to the "distance-decay" exponential models, while the vehicle categories were
consistently stable (Fig. 6). This demonstrates that the traffic fluxes played a cardinal role in determining
the spatial emission hotspot patterns over the highways. In contrast, in the arterial and residential roads,
coincident peaks of the specific vehicle categories (i.e., HDVs and HDTs) and traffic fluxes corresponded
in space to the emission hotspots (Fig. 6). This reveals that, besides the traffic fluxes, the specific vehicle
categories (i.e., HDVs and HDTs) also substantially contributed to the high emissions over the arterial
and residential roads.

## 3.5 Impacts of traffic control scenarios

Four scenarios (i.e., S1 ~ S4) demonstrated substantial impacts of traffic management on
spatiotemporal traffic states (Fig. S11). Therein the S1 and S2 scenarios focused on reducing the traffic





fluxes, while the S3 and S4 scenarios gave full consideration to not only the traffic fluxes but also the fleet compositions (Table 1). Without strict traffic restrictions, the S1 scenario only reduced the traffic fluxes by 3.82 % and showed no significant effects on the traffic flux hotspots. On this basis, the traffic fluxes were further decreased by 9.56 % under the S2 scenario. Most of the traffic flux hotspots vanished.

Under the S3 scenario, the fleet compositions were changed evidently over the highways, where HDTs and HDVs were removed. The S4 scenario combined the exclusive settings in the S2 and S3 scenarios and thus realized their consequences. This scenario achieved further decreases in the traffic fluxes (51.3%).

This study estimated that the daily average on-road vehicle emissions were 3.40 tons for CO,

0.482 tons for HC, 2.306 tons for $NO_x$, and 0.097 tons for $PM_{2.5}$ (Fig. 7 and Fig. S12). Under the S1 scenario, the daily average emissions decreased by small percentages (i.e., 3.74% for CO, 3.43% for HC, 3.13% for $NO_x$, and 3.08% for $PM_{2.5}$). Compared with the S1 scenario, the S2 scenario led to significant reductions over the arterial and residential roads (i.e., 9.34% for CO, 8.59% for HC, 7.83% for $NO_x$, and 7.69% for $PM_{2.5}$). This represents the effects of strictly traffic flux controls on on-road

vehicle emissions, particularly over the urban zones. The S3 scenario reveals that the HDVs and HDTs were responsible for a major part (15.94 ~ 57.50%) of CO, $NO_x$, $PM_{2.5}$, and HC emissions over the highways. On this basis, the S4 scenario adopted comprehensive traffic controls, thus reducing roughly additional emissions (i.e., 50.77% for CO, 67.44% for HC, 82.72% for $NO_x$, and 84.20% for $PM_{2.5}$). As a result, the emission hotspots over all roads were mostly removed.

**3.6 Comparison with other inventories**

The bottom-up on-road vehicle emissions in this study were compared with conventional emission inventories (i.e., MEICv1.3 for 2016 and HTAPv2.2 for 2010) (Fig. S13) (Janssens-Maenhout et al., 2015; Li et al., 2017). It should be noted that the spatial resolution of MEICv1.3 and HTAPv2.2 was roughly 0.25° * 0.25° and 0.1° * 0.1°, respectively. Hence, the total emissions over the Xiaoshan District were re-

aggregated with area-weighting. It was clear that the monthly average on-road vehicle emissions in this study were significantly lower than those in MEICv1.3 and HTAPv2.2 (i.e., MEIC:14.8% for CO, 30.1% for HC, 40.1% for $NO_x$, and 19.7% for $PM_{2.5}$. HTAP: 22.4% for CO, 44.5% for HC, 67.7% for $NO_x$ and





29.1% for PM$_{2.5}$). This could be attributed to the on-road vehicle emission mitigation measures in China. Moreover, owing to the limitation of the spatial resolution, these conventional inventories were incapable

of depicting the emission hotspots over the Xiaoshan District. Such limitations could be propagated to previous CTM simulations driven by those conventional inventories, thus unable to reproduce the fine-scale on-road vehicle emissions.

### 3.7 Stability Analysis

Through systematic subsampling of our weekday emission dataset, we found that 15 ~ 30

weekdays were sufficient to reproduce key spatial patterns with good precision and low bias (Figure 8). The following trends hold: a small number of drive days (N < 5) typically resulted in a poor approximation of long-term spatial patterns from the full data set, with generally low precision ($\boldsymbol{\gamma^2}$) and high bias (**CV-RMSE**). However, each additional sampling day resulted in a substantial improvement in $\boldsymbol{\gamma^2}$ and **CV-RMSE**.

For our dataset, diminishing returns for improvement in $\boldsymbol{\gamma^2}$ set in at 15 ~ 25 drive days, with mean $\boldsymbol{\gamma^2}$ for NO$_x$ and PM$_{2.5}$ approaching 0.7 after 15 weekdays, and approaching 0.9 after 30 ~ 35 weekdays. We found that ICC values ranged from 0.72 to 0.91 for all pollutants (Table S2). This indicates that our measurement-based long-term spatial patterns were robust to stochastic variability among those samples. Average values of ICC (Figure S14) were generally high for 10 or fewer weekdays, indicating that

stochastic temporal variability from a small number of weekdays did not obscure an overall spatially dominated emission pattern. Note that our sampling was restricted to weekday conditions, while spatial patterns may differ at other times (weekends) due to casual trips. Future work may reveal whether similar scaling considerations hold over a broader range of conditions.

### 4 Conclusions

In this study, comprehensive traffic conditions are fully measured and interconnected via the ITS. We find the spatial variabilities existing at much finer scales (i.e., 10m ~ 1km). Further, we capture the emission hotspots generally driven by high traffic fluxes. Over the highways, they are also associated with the distributions of HDVs and HDTs. Such findings might be missed by conventionally sparse





sampling and previous model predictions.  Overall, routine accessibility of hyperfine-resolution on-road

vehicle emissions could have transformative implications for air pollution control, urban management

and policymaking, epidemiology, and public awareness (Daellenbach et al., 2020; Dedoussi et al., 2020;

Geng et al., 2019; Nyhan et al., 2016; Yang and Zhang, 2018; Zeger et al., 2000). By pinpointing localized

emission hotspots, these data may provide new opportunities for air pollution control. In turn, Street-level

emission data can complement, challenge, and validate other emission and air quality datasets, including

CTM outputs and near-road air quality measurements (Apte et al., 2017; Grange et al., 2017; Jiang et al.,

2018; Yang et al., 2018a). Moreover, such refinements can help address exposure misclassification and

even directly alter personal behaviour, much as real-time traffic navigation data now inform individual

driving patterns. In addition, this hyperfine-resolution on-road vehicle emissions and subsequent air

quality maps might result in broader societal consequences, including urban land-use decisions,

ecological planning, and political economy.



*Data availability.* Traffic monitoring data and model results are available upon request.

*Supplement.* The supplement related to this article is available online.


*Author contributions.* S.Y., P.L. conceived and designed the research. L. J. performed model simulations. Y. X., X. C., L. W., and J. Y. conducted data analysis. T. H., L. W., Y. Z., M. L., Z. L., Z. S., Y. J., W. L., D. R., and J. H. S contributed to the scientific discussions. S.Y., P.L., and J. H. S wrote and revised the manuscript.

*Competing interests.* The authors declare that they have no conflict of interest.

*Acknowledgements.* This study is supported by the Department of Science and Technology of China (No. 2016YFC0202702, 2018YFC0213506 and 2018YFC0213503), National Research Program for Key Issues in Air Pollution Control in China (No. DQGG0107) and National Natural Science Foundation of China (No. 21577126 and 41561144004). Pengfei Li is supported
by Initiation Fund for Introducing Talents of Hebei Agricultural University (412201904), Hebei Youth Top Fund (BJ2020032), and National Natural Science Foundation of China (No. 22006030).

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

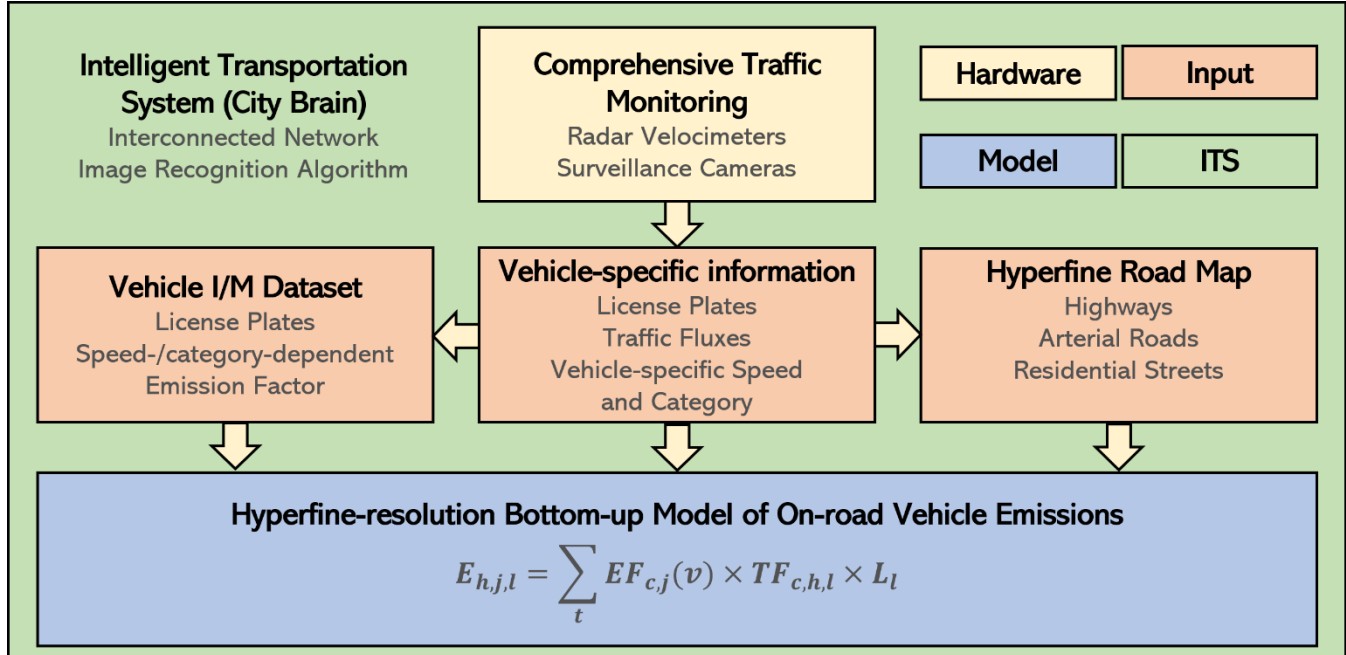

**Figure 1. A hyperfine-resolution bottom-up model framework for on-road vehicle emissions.** Each set of comprehensive traffic monitoring includes radar velocimeters and surveillance cameras. Consequently, vehicle-specific information is collected, including license plates, speed, categories, and traffic fluxes. According to the

license plates, the speed-/category-dependent emission factors are obtained from the local official vehicle I/M dataset. Over the entire district, road segments are divided into three road classes: highways, arterial roads, and residential streets. Each road segment is adaptive to a set of traffic monitoring that can collect comprehensive traffic profiles. Therefore, all-round traffic monitoring produces the hyperfine-resolution road map. On the basis of these input data, a hyperfine-resolution bottom-up model framework is established to calculate primary on-road vehicle

emissions. The detailed information is illustrated in Sect. 2.2. Here, an intelligent transportation system (ITS) (named "City Brain") is developed to interconnect these input data. In addition, an image recognition algorithm is embedded to recognize the category for a certain vehicle.

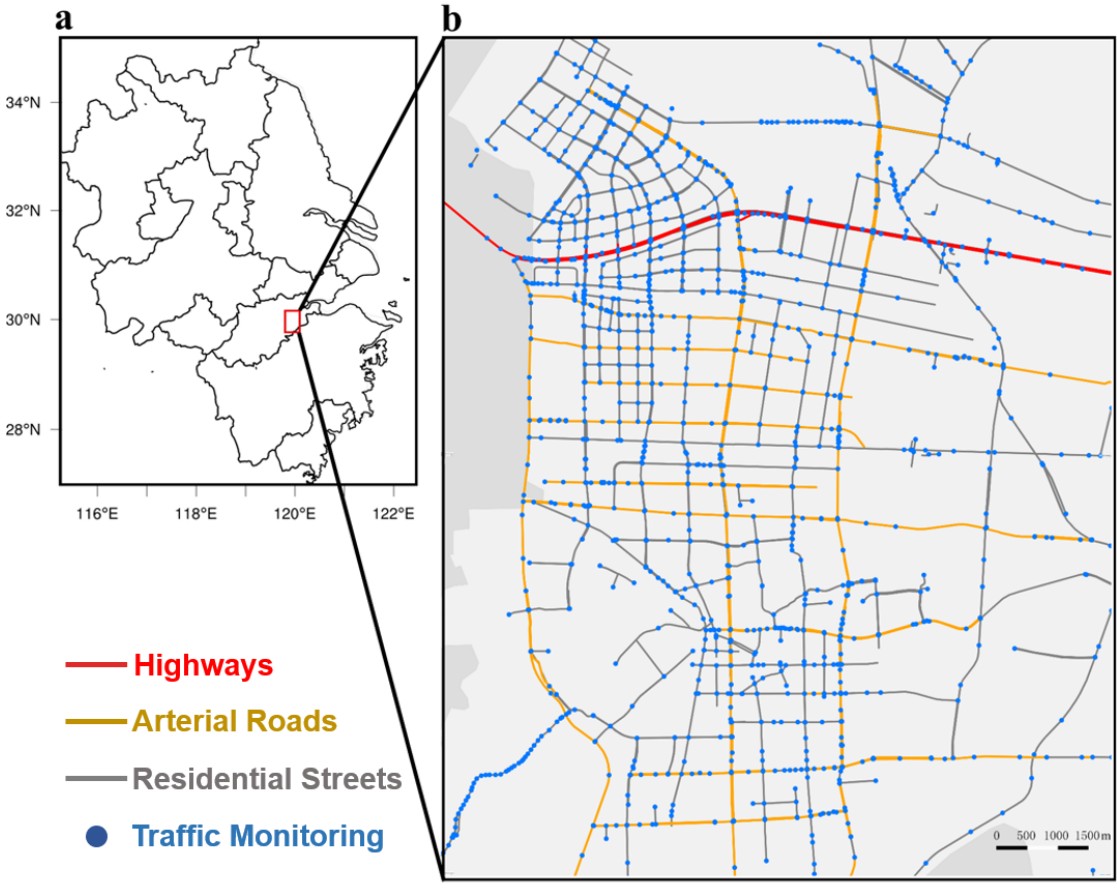

**Figure 2. Comprehensive traffic monitoring network in the Xiaoshan District.** (**a**) The Xiaoshan District (the red rectangle) is located in the hinterland of the YRD in China. (**b**) Comprehensive traffic monitoring achieves full coverage over the Xiaoshan District. Each dot represents a set of comprehensive traffic monitoring that can recognize traffic fluxes, vehicle-specific speed, categories, and license plates. The gaps between two sets, ranging from 10 m to 1 km, determine the spatial resolution of the road and emission map. The entire road network over the Xiaoshan District is divided into 1894 road segments. Such road segments are divided into three road classes: highways (red lines), arterial roads (yellow lines), and residential streets (grey lines). Map data © 2021, Gaode Map.




**Figure 3. Hyperfine-resolution mapping of observed traffic fluxes.** (**a**) Hourly average traffic fluxes in each road segment based on two-month traffic monitoring data over the entire district and (**b**) for three indicative 1 km$^2$





urban zones therein. The subgraphs (**c**) and (**d**) are as the same as (**a**) and (**b**) but for the hourly average traffic fluxes during the morning and evening rush hours (from 7:00 to 9:00 and from 16:30 to 18:30, Local Time) on weekdays (from Monday to Friday). The subgraphs (**e**) and (**f**) are as the same as (**a**) and (**b**) but for the hourly

average traffic fluxes of HDVs and HDTs. The subgraphs (**g**) and (**h**) are as the same as (**c**) and (**f**) but for the morning and evening rush hours on weekdays. The indicative zones are marked by white rectangles in (**a**) and their spatial scales are presented in (**b**). Illustrative traffic hotspots (circles) are also annotated in (**a**), including Tonghui North Road, Hongda Road, Shixin North Road, Shanyin Road, and Airport Road. Map data © 2021, Gaode Map.



**a**

**b**

**c**

**d**

**e**

**f**

**Figure 4. Imagery analysis for illustrative traffic hotspots. (a ~ b)** Over the hotspots in the urban zones, frequent large traffic fluxes are identified in major arterial roads and their intersections. (**c**) Constructions in the middle of the roads also lead to traffic congestion. (**d ~ e**) Morning and afternoon traffic rushes further deteriorate traffic congestion. (**f**) Over the highways, the hotspots are related to large traffic fluxes of MDTs and HDTs. The vehicle licence plates are pixelated. The hotspot locations are presented in Fig. 3.












**Figure 5. Hyperfine-resolution mapping of on-road vehicle emissions.** (**a**) Hourly average on-road vehicle $NO_x$ emissions for each road segment based on two-month traffic monitoring data for the entire district and (**b**) for three indicative 2.5 km² urban zones therein. The subgraphs (**c**) and (**d**) are as the same as (**a**) and (**b**) but for the hourly average emissions during the morning and evening rush hours (from 7:00 to 9:00 and from 16:30 to 18:30, Local Time) on weekdays (from Monday to Friday). The subgraphs (**e**) and (**f**) are as the same as (**a**) and (**b**) but for the hourly average emissions of HDTs and HDVs. The subgraphs (**g**) and (**h**) are as the same as (**a**) and (**b**) but for the hourly average emissions of HDTs and HDVs during the morning and evening rush hours on weekdays. The indicative zones are marked by white rectangles in (**a**). Map data © 2021, Gaode Map.








**Figure 6. Decay of on-road vehicle emissions, traffic fluxes, and vehicle categories from indicative hotspots outwards on weekdays.** (**a ~ d**) Points denote the ratio of hourly average emissions (NO$_x$, CO, HC, and PM$_{2.5}$) over different road types at a given distance from hotspots to hourly average hotspot emissions. Error bars present standard deviations. An unconstrained three parameter exponential model reproduces the decay relationships with high fidelity. The isotropic parameter **d** reflected the distance to the hotspot cores (**m**); the background parameter **α** represented the background emissions far from the hotspots (**d → 1000m**); the parameter **β** represented the emission increment resulting from proximity to the hotspots; the decay parameter **k** governed the spatial scale over which emissions relaxed to **α**. The subgraphs (**e**) and (**f**) are the same as (**a**) but for traffic fluxes and the vehicle category proportions, respectively.






**Figure 7. Impacts of traffic control strategies (i.e., S1, ~ S4) on daily average on-road vehicle NOₓ emissions.**

Map data © 2021, Gaode Map.

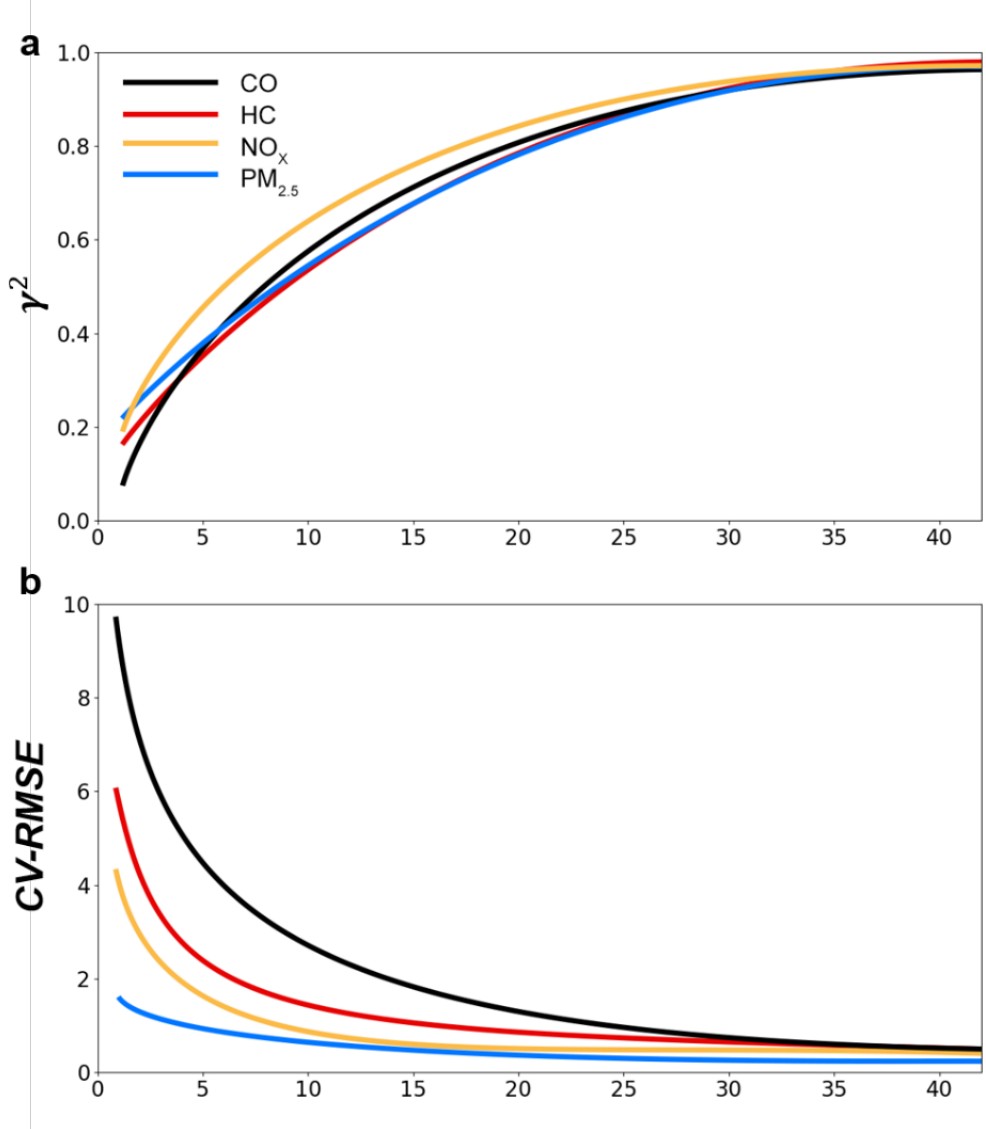

**Figure 8. Scaling analysis through systematic subsampling. (a)** Mean subsampled $\gamma^2$ as a function of each road segment emissions relative to the full data set, plotted as function the number of unique weekdays for CO, HC, $NO_x$, and $PM_{2.5}$. **(b)** Mean subsampled coefficient of variation of root mean squared errors (**CV-RMSE**) versus the number of unique weekdays for CO, HC, $NO_x$, and $PM_{2.5}$.






**Table 1. Traffic control strategies.** The detailed rules are presented spatially and temporally.

| Traffic control strategy | Time scale | Space scale | Vehicle category | Rule |
|---|---|---|---|---|
| S1 | Morning and evening rush hours during from Monday to Friday | Arterial and residential roads | All | For each weekday (from Monday to Friday), vehicles with specific tail numbers of the license plates were prohibited on the arterial and residential roads during the morning and evening rush hours (from 7:00 to 9:00 and from 16:30 to 18:30, Local Time). The prohibited tail numbers were 1 and 9 on Monday, 2 and 8 on Tuesday, 3 and 7 on Wednesday, 4 and 6 on Thursday, and 5 and 0 on Friday. |
| S2 | Morning and evening rush hours during from Monday to Friday | Arterial and residential roads | All | The even-odd rule for the license plates was adopted over the arterial and residential roads on weekdays. |
| S3 | All day | Highways | HDVs and HDTs | Both local registered and non-registered trucks were strictly prohibited all day long over the highways. |
| S4 | All day | All roads | All | All kinds of vehicles complied with the even-odd rule of the license plates over the entire District. |