# Peer review of "Hyperfine-Resolution Mapping of On-Road Vehicle Emissions with Comprehensive Traffic Monitoring and Intelligent Transportation System"

_Atmospheric Chemistry and Physics, 2021_

## Referee Comment (RC1)

**Comments on "Hyperfine-Resolution Mapping of On-Road Vehicle Emissions with Comprehensive TrafficMonitoring and Intelligent Transportation System" by *Linhui Jiang et al., Atmos. Chem. Phys. Discuss., https://doi.org/ 10.5194/acp -2021-533, 2021***

The manuscript by Jiang et al. establishes an urban on-road vehicle-specific emission inventory, which makes an important technological breakthrough and is central to urban ozone and particular matter pollution control.In particular, this study proposes a hyperfine-resolution bottom-up model framework built upon a series of valuable ITS facilities and algorithms, like radar velocimeters, surveillance cameras, and the image recognition algorithm.To my knowledge, this is the first time for investigating urban on-road emissions with a resolution up to several meters. The authors much have gone a long way towards such findings.Consequently,an unprecedentedemission map is obtained in this study. Therein, widespread and persistent emission hotspots emerged. They are of significantly sharp small-scale variability, up to 8 ~ 15 times within individual hotspots, attributable to distinct traffic fluxes, road conditions, and vehicle categories.

Overall, this work is novel, important, and well-written. I recommend its acceptance for publication afterminor revisions.

General Comments:

Line 394. The authors conclude with severalfinal important implications of this work. For the public, as pointed by the authors, "the hyperfine-resolution emission inventory can alter personal behaviour, much as real-time traffic navigation data now inform individualdriving patterns.". However, policymakers still question how this hyperfine-resolution emission inventory would help improve air quality and address exposure misclassification.Hence, the discussions would be more insightful if the authors could make this clearer. I believethe chemical transport model (CTM) might be a key link.

Specific comments:

Line 113:What is the exact period of the "rush hours"? As appearing for the first time,it has to be specified.

Line 171:Does "during the morning and evening rush hours" mean the same as "during the morning and afternoon rush hours" (Line 113)?If so, please unify the definitions.

Line 159: Please give brief definitions for the "light-duty vehicles (LDVs), middle-duty vehicles(MDVs), heavy-duty vehicles (HDVs), light-duty trucks (LDTs), middle-duty trucks (MDTs), and heavy-duty trucks (HDTs)?"

Line 222: The "overall" should be deleted.

Figure 8: This picture lacks the description of the abscissa.Is that the number of weekdays?

---

## Author Comment (AC3)

**Reply to comments on "Hyperfine-Resolution Mapping of On-Road Vehicle Emissions with Comprehensive Traffic Monitoring and Intelligent Transportation System" by Linhui Jiang et al.**

**Reply to Reviewer #1**

The manuscript by Jiang et al. establishes an urban on-road vehicle-specific emission inventory, which makes an important technological breakthrough and is central to urban ozone and particular matter pollution control. In particular, this study proposes a hyperfine resolution bottom-up model framework built upon a series of valuable ITS facilities and algorithms, like radar velocimeters, surveillance cameras, and the image recognition algorithm. To my knowledge, this is the first time for investigating urban on-road emissions with a resolution up to several meters. The authors much have gone a long way towards such findings. Consequently, an unprecedented emission map is obtained in this study. Therein, widespread and persistent emission hotspots emerged. They are of significantly sharp small-scale variability, up to 8 ~ 15 times within individual hotspots, attributable to distinct traffic fluxes, road conditions, and vehicle categories. Overall, this work is novel, important, and well-written. I recommend its acceptance for publication after minor revisions.

**Response: We truly appreciate the interest and support of the reviewer. We are also grateful for all the constructive comments and suggestions. We have adopted all the suggestions in our revised manuscript.**

**The followings are our point-to-point responses to the reviewers' comments. The responses are shown in brown and bold fonts, and the added/rewritten parts for the revision are presented in blue and bold fonts.**

**General comments:**

Line 394. The authors conclude with several final important implications of this work. For the public, as pointed by the authors, "the hyperfine-resolution emission inventory can alter personal behaviour, much as real-time traffic navigation data now inform individual driving patterns.". However, policymakers still question how this hyperfine resolution emission inventory would help improve air quality and address exposure misclassification. Hence, the discussions would be more insightful if the authors could make this clearer. I believe the chemical transport model (CTM) might be a key link.

**Response: We thank the reviewer for raising this important issue. We have supplemented associated discussions to clarify this issue.**

**Added/rewritten part in Conclusions:** By pinpointing localized emission hotspots, these data may provide new opportunities for policymakers. Specifically, our results can replace the coarse-grid ($> 1 \times 1 \sim 25 \times 25$ km$^2$) emission inventory (Janssens-Maenhout et al., 2015; Li et al., 2017; Zhang et al., 2013) as the input for the CTM. Comparably, the meteorological input should also be hyperfine sufficiently, which thus needs to account for large eddy simulations (e.g., WRF-LES). In so

doing, dispersion models (e.g., AERMOD) (Yang et al., 2019), instead of full CTMs (Mehmood et al., 2020; Wong et al., 2012; Yu et al., 2013), are sufficient to resolve street-level gradients of air pollution concentrations. Through combination with CTM outputs and near-road air quality measurements (Apte et al., 2017; Grange et al., 2017; Jiang et al., 2018; Yang et al., 2018), the hyperfine-resolution scanning of responses of air quality to emissions becomes possible. This would help understand highly nonlinear air pollution mechanisms, such as the $O_3$-VOCs-$NO_x$ relationships (Li et al., 2019), and thus optimize mitigation policies. Besides, the resulting hyperfine-resolution maps of air pollutant concentrations can help address exposure misclassifications and even directly alter personal behaviours, such as real-time traffic navigation data can now inform individual driving patterns. In addition, these hyperfine-resolution emissions and air quality maps might result in broader societal consequences, including urban land-use decisions, ecological planning, and political economy.

**Specific comments:**

Line 113: What is the exact period of the "rush hours"? As appearing for the first time, it has to be specified.

**Response: We thank the reviewer for the suggestion. We have supplemented the exact period of the "rush hours"?**

**Added/rewritten part in Comprehensive traffic monitoring network:** Since 2016, routine measures to ease traffic congestions, such as license restrictions during the morning and evening rush hours (from 7:00 to 9:00 and from 16:30 to 18:30, Local Time) on weekdays (i.e., from Monday to Friday), were implemented over the Xiaoshan District.

Line 171: Does "during the morning and evening rush hours" mean the same as "during the morning and afternoon rush hours" (Line 113)? If so, please unify the definitions.

**Response: We thank the reviewer for the suggestion. We have unified the definitions.**

Line 159: Please give brief definitions for the "light-duty vehicles (LDVs), middle-duty vehicles (MDVs), heavy-duty vehicles (HDVs), light-duty trucks (LDTs), middle-duty trucks (MDTs), and heavy-duty trucks (HDTs)?"

**Response: We thank the reviewer for the suggestion. We have supplemented the definitions. This classification follows the national standard. The LDVs were all designated as vehicles in a length of ≤6 m and ridership of ≤9. The MDVs and HDVs were of the same length but with different ridership of 10 ~ 19 and >20, respectively. More definitions of the trucks could be found in the national standard profile.**

**Added/rewritten part in Hyperfine-resolution bottom-up model framework:** Herein, six vehicle categories were detected and defined, including light-duty vehicles (LDVs), middle-duty vehicles (MDVs), heavy-duty vehicles (HDVs), light-duty trucks (LDTs), middle-duty trucks (MDTs), and heavy-duty trucks (HDTs). This classification follows the national standard (GA802-2008). The LDVs were all designated as vehicles in a length of ≤6 m and ridership of ≤9. The MDVs and HDVs were of the same length but with different ridership of 10 ~ 19 and >20, respectively. More definitions of the trucks could be found in the national standard profile (GA802-2008).

Line 222: The "overall" should be deleted.

**Response: We thank the reviewer for the suggestion. We have deleted the word.**

70  Figure 8: This picture lacks the description of the abscissa. Is that the number of weekdays?

**Response: We thank the reviewer for the suggestion. We have supplemented the abscissa. We have also revised the figure caption to make it clearer.**

75

---

## Author Comment (AC4)

**Reply to comments on "Hyperfine-Resolution Mapping of On-Road Vehicle Emissions with Comprehensive Traffic Monitoring and Intelligent Transportation System" by Linhui Jiang et al.**

5 **Reply to Reviewer #2**

This paper established a high spatial resolution bottom-up on-road vehicle emission inventory using measured traffic fluxes, vehicle-specific categories, and speeds over the Xiaoshan District in the Yangtze River Delta (YRD) region. The effectiveness of traffic control strategies was investigated based on the hyperfine on-road vehicle emission dataset.

10 The importance of controlling the mobile sources on the synergy effect of PM2.5 and O3 abatement draws more and more attentions in recent years. However, the uncertainties in current mobile source emission inventory propagates large biases to the model simulation results and final control measures development. As a modeler, I am excited to see that using on-site measurement and big data technology is able to establish such fine resolution mobile source emission inventory. It can also improve the accuracy of model simulations significantly.

15 This paper is good in general and within the scope of Atmospheric Chemistry and Physics. I recommend for publication once the comments expressed below are addressed.

**Response: We truly appreciate the interest and support of the reviewer. We are also grateful for all the constructive comments and suggestions. We have adopted most of the suggestions in our revised manuscript.**

**The followings are our point-to-point responses to the reviewers' comments. The responses are shown in brown**
20 **and bold fonts, and the added/rewritten parts are presented in blue and bold fonts.**

**General comments:**

1. The author needs to add some discussions regarding how to put this hyperfine on-road vehicle emission into the air quality models. Is it feasible and cost effective to build a nation-wide hyperfine on-road vehicle emission using the same method
25 established in this study?

**Response: We thank the reviewer for the suggestion. We have supplemented associated discussions to clarify how to put the resulting hyperfine emission inventory into the CTMs. Specifically, our results can replace the coarse-grid ($> 1 \times 1 \sim 25 \times 25$ km$^2$) emission inventory (Janssens-Maenhout et al., 2015; Li et al., 2017; Zhang et al., 2013) as the input of the CTM. Comparably, the meteorological input should also be hyperfine sufficiently, which thus needs to account**
30 **for large eddy simulations (e.g., WRF-LES). In so doing, dispersion models (e.g., AERMOD) (Yang et al., 2019), instead of full CTMs (Mehmood et al., 2020; Wong et al., 2012; Yu et al., 2013), are sufficient to resolve street-level gradients of air pollution concentrations. Through combination with CTM outputs and near-road air quality measurements**

(Apte et al., 2017; Grange et al., 2017; Jiang et al., 2018; Yang et al., 2018), the hyperfine-resolution scanning of responses of air quality to emissions becomes possible. This would help understand highly nonlinear air pollution mechanisms, such as the $O_3$-VOCs-$NO_x$ relationships (Li et al., 2019), and thus optimize mitigation policies. Besides, the resulting hyperfine-resolution map of air pollutant concentrations can help address exposure misclassification and even directly alter personal behaviour, such that real-time traffic navigation data now inform individual driving patterns. In addition, these hyperfine-resolution emissions and air quality maps might result in broader societal consequences, including urban land-use decisions, ecological planning, and political economy.

Besides, we would note that this type of hyperfine on-road vehicle emission inventories can be established in our way nationwide. However, it should be conducted strategically due to high costs. We have supplemented detailed discussions on this issue.

This work proposes a straightforward emission model framework that can provide several orders of magnitude more spatial information. As shown in our results, this approach could be extended to nationwide megacities if comprehensive traffic conditions are fully measured and interconnected via the ITS. However, its costs are significantly higher than those of previous attempts. To this end, more flexible data collection from low-cost sensors, such as those on cell phones, taxis, and public transit, could substantially lower the costs of monitoring instruments. Furthermore, advances in open-source traffic platforms that can complete those big data interconnections would further decrease the costs. In addition, as demonstrated in Sect. 3.7, our approach, coupled with data reduction algorithms, might also enable high-resolution emission mappings. This indicates an application potential of our approach for middle-sized and small cities where robust traffic monitoring infrastructures are absent.

**Added/rewritten part in Conclusions:** By pinpointing localized emission hotspots, these data may provide new opportunities for policymakers. Specifically, our results can replace the coarse-grid ($> 1 \times 1 \sim 25 \times 25 \ km^2$) emission inventory (Janssens-Maenhout et al., 2015; Li et al., 2017; Zhang et al., 2013) as the input of the CTM. Comparably, the meteorological input should also be hyperfine sufficiently, which thus needs to account for large eddy simulations (e.g., WRF-LES) (Zhong et al., 2020). In so doing, dispersion models (e.g., AERMOD) (Yang et al., 2019), instead of full CTMs (Mehmood et al., 2020; Wong et al., 2012; Yu et al., 2013), are sufficient to resolve street-level gradients of air pollution concentrations. Through combination with CTM outputs and near-road air quality measurements (Apte et al., 2017; Grange et al., 2017; Jiang et al., 2018; Yang et al., 2018), the hyperfine-resolution scanning of responses of air quality to emissions becomes possible. This would help understand highly nonlinear air pollution mechanisms, such as the $O_3$-VOCs-$NO_x$ relationships (Li et al., 2019), and thus optimize mitigation policies. Besides, the resulting hyperfine-resolution maps of air pollutant concentrations can help address exposure misclassifications and even directly alter personal behaviours, such that real-time traffic navigation data can now inform individual driving patterns. In addition, these hyperfine-resolution emission and air quality maps might result in broader societal consequences, including urban land-use decisions, ecological planning, and political economy.

**Added/rewritten part in Conclusions:** This work proposes a straightforward emission model framework that can provide several orders of magnitude more spatial information. As shown in our results, this approach could be extended to

nationwide megacities if comprehensive traffic conditions are fully measured and interconnected via the ITS. However, its costs are significantly higher than those of previous attempts. To this end, more flexible data collection from low-cost sensors, such as those on cell phones, taxis, and public transit, could substantially lower the costs of monitoring instruments. Furthermore, advances in open-source traffic platforms that can complete those big data interconnections would further decrease the costs. In addition, as demonstrated in Sect. 3.7, our approach, coupled with data reduction algorithms, might also enable high-resolution emission mappings. This indicates an application potential of our approach for middle-sized and small cities where robust traffic monitoring infrastructures are absent.

2. The author needs to add some discussions on the uncertainties of the hyperfine on-road vehicle emission established in this study. It seems that the vehicle emission activities can be greatly improved, what about emission factors. The method used in this study divides vehicles into 6 categories. Does it include and separate gasoline and diesel vehicles, and does it take vehicle age into account?

**Response: We thank the reviewer for the suggestion. We have supplemented associated discussions for the uncertainties in our results. In particular, the uncertainties in emission factors have been involved. Also, the fuel- and age- associated uncertainties are discussed briefly.**

**As pointed by the reviewer, in our model framework, the traffic fluxes are measured accurately. By comparison, the emission factors are of larger uncertainties. This is because although they are obtained from the local official vehicle Inspect/Maintenance (I/M) datasets, some assumptions are inappropriate. For instance, the emission factors are measured in lab circumstances, possibly unsuitable for real-world conditions (Seo et al., 2021). Besides, they are calculated as a function of the vehicle categories and speeds (Fig. S1), without consideration of fuel-dependent discrepancies. Instead, we assumed that, in this study, HDVs and HDTs are diesel-driven, while other vehicle categories are fueled by gasoline. Also, the effects of vehicle ages were ignored. Such assumptions are consistent with previous studies (Yang et al., 2019; Zhou et al., 2017). Future introductions of constraints via near-road emission measurements would decrease such uncertainties.**

**Added/rewritten part in Conclusions:** In our model framework, the traffic fluxes are measured accurately. By comparison, the emission factors are of larger uncertainties. This is because although they are obtained from the local official vehicle Inspect/Maintenance (I/M) datasets, some assumptions are inappropriate. For instance, the emission factors are measured in lab circumstances, possibly unsuitable for real-world conditions (Seo et al., 2021). Besides, they are calculated as a function of the vehicle categories and speeds (Fig. S1), without consideration of fuel-dependent discrepancies. Instead, we assumed that, in this study, HDVs and HDTs are diesel-driven, while other vehicle categories are fueled by gasoline. Also, the effects of vehicle ages were ignored. Such assumptions are consistent with previous studies (Yang et al., 2019; Zhou et al., 2017). Future introductions of constraints via near-road emission measurements would decrease such uncertainties.

**Specific comments:**

1. In section 3.6, the author compared the newly established on-road vehicle emission inventory with those from MEIC and HTAP inventories at regional scale. Is it possible to add some comparisons with localized refined emission inventory, as well as with the measured vehicle emission factors in the literature?

**Response: We thank the reviewer for the suggestion. We have supplemented associated discussions for this issue. In this study, the emission factors were measured. We obtained them from the local official vehicle Inspect/Maintenance (I/M) datasets, the methodology of which was described in China's National Emission Inventory Guidebook (ICCT, 2020). On the other hand, the localized emission inventory over the Xiaoshan District is still lacking. MEICv1.3 for 2016 and HTAPv2.2 for 2010, as state-of-the-art conventional emission inventories provide a valuable opportunity to evaluate our results (Fig. S13) (Janssens-Maenhout et al., 2015; Li et al., 2017).**

**Added/rewritten part in Comparison with other inventories:** The localized emission inventory over the Xiaoshan District is still lacking. MEICv1.3 for 2016 and HTAPv2.2 for 2010, as state-of-the-art conventional emission inventories provide a valuable opportunity to evaluate our results (Fig. S13) (Janssens-Maenhout et al., 2015; Li et al., 2017).

2. The caption in Figure 1 needs to be simplified, no necessary to explain the method again.

**Response: We thank the reviewer for the suggestion. We have simplified the caption accordingly.**

**Added/rewritten part in Figure 1: Figure 1. A hyperfine-resolution model framework for on-road vehicle emissions.** Traffic monitoring includes radar velocimeters and surveillance cameras. License plates, speeds, categories, and traffic fluxes are collected. The speed-/category-dependent emission factors are obtained from the local official vehicle I/M datasets. Road segments are divided into three road classes: highways, arterial roads, and residential streets. An intelligent transportation system (ITS) (named "City Brain") is developed to interconnect these input data. An image recognition algorithm is embedded to recognize the category for a certain vehicle. The detail information is illustrated in Sect. 2.2.

3. In the line 211, the author takes ICC values of 0.75 ~ 1 as the reflection of large and systematic spatial differences. What is the basis for this range? More explanation is needed.

**Response: We thank the reviewer for the suggestion. We have supplemented associated interpretations and references for the application of the ICC. ICC is a common evaluation parameters in intra- and inter-rater reliability analyses (Bartko, 1966; Koo and Li, 2016; Shrout and Fleiss, 1979). By definition, a low ICC can relate to the lack of variability among sampled subjects, while a high value indicates that substantially more variabilities occur among groups than does within each group. For a hypothetical dataset where all repeated measurements at each location were precisely equal to each other, the ICC would converge to 1.0. In contrast, for a dataset where the concentration variabilities among repeated measures at each individual location are very high relative to the spatial differences in concentrations among roads, the ICC would approach 0. Previous studies suggest that ICC values less than 0.5 are indicative of poor reliability, values between 0.5 and 0.75 indicate moderate reliability, values larger than 0.75**

**indicate good reliability (Bartko, 1966; Koo and Li, 2016; Shrout and Fleiss, 1979). For this application, ICC values**

135 **of 0.75 ~ 1 reflected large and systematic spatial differences, with a low residual temporal variability at each location.**

**Added/rewritten part in Monte Carlo subsampling:** ICC is a common evaluation parameters in intra- and inter-rater reliability analyses (Bartko, 1966; Koo and Li, 2016; Shrout and Fleiss, 1979). By definition, a low ICC can relate to the lack of variabilities among sampled subjects, while a high value indicates that substantially more variabilities occur among groups than does within each group. For a hypothetical dataset where all repeated measurements at each location were

140 precisely equal to each other, the ICC would converge to 1.0. In contrast, for a dataset where the concentration variabilities among repeated measures at each individual location are very high relative to the spatial differences in concentration among roads, the ICC would approach 0. Previous studies suggest that ICC values less than 0.5 are indicative of poor reliability, values between 0.5 and 0.75 indicate moderate reliability, values larger than 0.75 indicate good reliability (Bartko, 1966; Koo and Li, 2016; Shrout and Fleiss, 1979). For this application, ICC values of 0.75 ~ 1 reflected large and systematic

145 spatial differences, with a low residual temporal variability at each location.

4. More quantitative findings from this study need to add into conclusion part in section 4.

**Response: We thank the reviewer for the suggestion. We have revised this paragraph to make it more quantitative and illustrative.**

150 **Added/rewritten part in Conclusions:** This work establishes a hyperfine bottom-up approach to reveal a unique on-road vehicle emission pattern at 1 ~ 3 orders of magnitude higher spatial resolution than current emission inventories. In particular, all-around traffic monitoring (including traffic fluxes, vehicle-specific categories, and speeds) is interconnected via an intelligent transportation system (ITS) over the Xiaoshan District in the Yangtze River Delta (YRD) region. This enables us to calculate single-vehicle-specific emissions over each fine-scale (10m ~ 1km) road segment. Consequently, the most

155 hyperfine emission dataset of its type is achieved, exposing widespread and persistent emission hotspots. More importantly, this map is of significantly sharp small-scale variabilities, up to 8 ~ 15 times within individual hotspots, attributable to distinct traffic fluxes, road conditions, and vehicle categories. Once all kinds of vehicles comply with the even-odd rule over the entire district, more than 50% of the emissions are reduced. By comparison, our results are lower ($>$ 14.8% ~ 67.7%) than those in the conventional emission inventories (i.e., MEICv1.3 and HTAPv2.2). Through systematic subsampling of our weekday

160 emission dataset, we find that 15 ~ 30 weekdays are sufficient to reproduce key spatial patterns with good precision and low bias.